# Renoprotective and haemodynamic effects of adiponectin and peroxisome proliferator-activated receptor agonist, pioglitazone, in renal vasculature of diabetic Spontaneously hypertensive rats

Sheryar Afzal[1,2]*, Munavvar Abdul Sattar[1,2ᵒ], Edward James Johns[3ᵒ], Olorunfemi A. Eseyin[1,4ᵒ]

1 School of Pharmaceutical Sciences, University Sains Malaysia, Penang, Malaysia, 2 Faculty of Pharmacy, MAHSA University, Selangor, Malaysia, 3 Faculty of Pharmacy, University of Uyo, Uyo, Akwa Ibom State, Nigeria, 4 Department of Physiology, University College Cork, Cork, Ireland

ᵒ These authors contributed equally to this work.
* samalik77@hotmail.com

**Data Availability Statement:** All relevant data are within the manuscript and its Supporting Information files.

## Abstract

Pioglitazone, a therapeutic drug for diabetes, possesses full PPAR-γ agonist activity and increase circulating adiponectin plasma concentration. Plasma adiponectin concentration decreases in hypertensive patients with renal dysfunctions. Present study investigated the reno-protective, altered excretory functions and renal haemodynamic responses to adrenergic agonists and ANG II following separate and combined therapy with pioglitazone in diabetic model of hypertensive rats. Pioglitazone was given orally [10mg/kg/day] for 28 days and adiponectin intraperitoneally [2.5μg/kg/day] for last 7 days. Groups of SHR received either pioglitazone or adiponectin in combination. A group of Wistar Kyoto rats [WKY] served as normotensive controls, whereas streptozotocin administered SHRs served as diabetic hypertensive rats. Metabolic data and plasma samples were taken on day 0, 8, 21 and 28. In acute studies, the renal vasoconstrictor actions of Angiotensin II [ANGII], noradrenaline [NA], phenylephrine [PE] and methoxamine [ME] were determined. Diabetic SHRs control had a higher basal mean arterial blood pressure than the WKY, lower RCBP and plasma adiponectin, higher creatinine clearance and urinary sodium excretion compared to WKY [all P<0.05] which were normalized by the individual drug treatments and to greater degree following combined treatment. Responses to intra-renal administration of NA, PE, ME and ANGII were larger in diabetic SHR than WKY and SHRs [P<0.05]. Adiponectin significantly blunted responses to NA, PE, ME and ANG II in diabetic treated SHRs by 40%, whereas the pioglitazone combined therapy with adiponectin further attenuated the responses to adrenergic agonists by 65%. [all P <0.05]. These findings suggest that adiponectin possesses renoprotective effects and improves renal haemodynamics through adiponectin receptors and PPAR-γ in diabetic SHRs, suggesting that synergism exists between adiponectin and pioglitazone. A cross-talk relationship also supposed to exists between adiponectin receptors, PPAR-γ and alpha adrenoceptors in renal vasculature of diabetic SHRs.

**Funding:** This study was funded by: the Universiti Sains Malaysia, Research grant # 1001/PFARMASI/815078. Sheryar Afzal is a recipient of USM Fellowship from the Institute of Post-Graduate Studies (IPS), Universiti Sains Malaysia (USM), Penang, Malaysia. No additional external funding was received for this study.

**Competing interests:** Authors have no competing interests for this manuscript

# Introduction

The worldwide prevalence and incidence of Diabetes Mellitus (DM) and hypertension is increasing alarmingly as the two most chronic non-communicable diseases [1, 2], whereas, also frequently coexist in individuals. Obesity and poor dietary choices serves as the mutual risk factors for their coexistence [3], therefore, combination treatment are often prescribed for individuals suffering from either of these ailments.

Pioglitazone, thiazolidinedione (TZDs) derivative is a hypoglycemic agents which acts as a ligand for nuclear peroxisome proliferator–activated receptors-γ (PPAR-γ) [4] in adipose tissue, and is used therapeutically to ameliorate insulin resistance through activation of PPAR-γ receptor, thus results in increased adiponectin levels and number of small adipocytes leading to enhanced insulin sensitivity [5], increased AMPK activation and lessening gluconeogenesis in liver [6]. Previous studies have revealed that TZDs, acting as full PPARγ agonist [7], possesses renoprotective effects, such as reduced albuminuria and glomerulopathy, in various renal disorders [8] including diabetic nephropathy, besides ameliorating hyperglycemia in diabtic individudlas [9]. In addition, pioglitazone has the potential of modulating blood pressure by causing reduction in sympathetic over activity, improving endothelial function [10], and reduction in vascular sensitivity to Ang II together with endothelial dys-function [11], through enhanced adiponectin concentration in plasma which in turn arouses the production of nitric oxide [12].

Adiponectin causes vasodilatation through stimulation of nitric oxide (NO) release from the vascular endothelium thereby causing vasodilation [13]. The chronic changes as causative to vascular diseases in diabetes and hypertension could also be inhibited by plasma adiponectin levels, which modulates by AT1 blockers [14]. In addition, plasma adiponectin concentration correlated with vasodilator response to reactive hyperemia suggests that hypoadiponectinemia associated with impaired endothelium-dependent vasorelaxation [15]. The direct effect of PPAR-γ on adiponectin transcription is through PPAR-γ activation [16] with a PPARγ- ligand which is useful for insulin-resistant patients with vascular disorders and hypertension [17]. Moreover, previous studies showed that decreased plasma adiponectin concentration is associated with hypertension and with renal dysfunction [18, 19].

In addition, the cross talk between adrenergic neurotransmission and Ang II in rats has been shown to play vital role in determining the tone of vasculature in various pathological states. Furthermore, evidence also exists for the interaction between α1-adrenergic receptors and Ang II receptors in vasculature of SHRs [20]. However, presently, there is no proof that TZDs and adiponectin interact pharmacodynamically in diabetic spontaneously hypertensive rats (SHRs).

In the light of the above background we aim to know that whether in hypertensive rat model (SHRs); adiponectin could decrease the blood pressure with reno-protective action of PPAR-γ agonist. The hypothesis explored was that in diabetic SHRs, circulating adiponectin levels would be decreased and with the separate administration of exogenous adiponectin and pioglitazone exogenously, would lower B.P and regulate renal haemodynamics response to vasoconstrictor agents to a greater extent following combined adiponectin and pioglitazone treatments. We also sought to categorize an interaction in vivo the subset of alpha-1-adrenoceptors (α1A, α1B and α1D), PPAR-γ with adiponectin receptors (adipo R1 & adipo R2) which that contribute to the renal vascular responses, altered metabolic and excretory functions of adiponectin in diabetic model of hypertensive rats.

# Material and methods

## Ethical approval for the study

This study was carried out in accordance with the recommendations and guidelines by the research center "Animal Research and Service Centre (ARASC), USM (Main Campus)" with

ethical approval number: 2012 (75) (352). The protocol for the study was approved by the "Animal Ethics Committee, Universiti Sains Malaysia, (AECUSM), Malaysia".

## Experimental animals

Thirty Spontaneously hypertensive rats (SHRs) and Six Wistar Kyoto (WKY) rats averaging body weight from (230-255g) were used in this study. All the animals were obtained from animal house facility of Universiti Sains Malaysia, Malaysia. Rats were observed for any form of the congenital abnormality and were declared fit to be used by the experts/veterinarian of the housing facility as per protocol and requirements of the experiment.

**Housing and husbandry.** Rats were housed in stainless metabolic cages with exposure to light during day and night hours in the animal transit room located in the school of Pharmaceutical Sciences, Universiti Sains Malaysia, Malaysia for 3 days to acclimatize to the environment before the experiment The maximum number of rats per housing cage was limited to 6 animals and the bottom of each housing cage was embedded with pine bedding (Living World® Pine Shavings, Hagen, Holland). The bedding was changed on every other day to optimize the normal physiological needs of the animals such as defecation and urination and temperature of the room was control using the thermometer. Commercial rat chow (Gold Coin Sdn. Bhd., Penang, Malaysia) and tap water using plastic bottles were provided ad-libitum to all experimental animals. The drinking bottles were cleaned daily to ensure fresh drinking water. The frequency of the animal monitoring was twice daily for any kind of untoward observation which could either affect experimental readings or lead to animal sufferings. None of the rats were killed or rejected during the whole experimental protocol. Any kind of the suffering or distress was avoided through continuous supervision of the animals, therefore, no analgesics were used during the experiment. All procedures and rats handling were carried out in accordance with guidelines of Animal Ethics Committee, Universiti Sains Malaysia, (AECUSM), Malaysia.

**Experimental grouping of rats.** Six groups of rats were studied, one of Wistar control animals and five of SHR [n = 6 in each group] treated as follows:

Wistar rats: treated with vehicle (WKY); SHR: treated with vehicle (SHR); SHR treated with streptozotocin (SHR+STZ)

SHR+STZ+Pio: given pioglitazone (10mg/kg) by oral gavage for 28 days starting from day 1,

SHR+STZ+Adp: given adiponectin 2.5μg/kg/day, intra-peritoneal, day 21 to day 28 [21, 22]

SHR+STZ+Pio+Adp: given pioglitazone (10mg/kg) by oral gavage for 28 days starting from day 1, and adiponectin 2.5μg/kg/day, intra-peritoneal, from day 21 to day 2

**Termination of the experiment.** At the end of the acute experiment, rats underwent surgical intervention were sacrificed with an overdose of sodium pentobarbitone, Nembutal®, CEVA, France) and disposed off in accordance with guidelines of Animal Ethics Committee of Universiti Sains Malaysia.

## Drugs

1. Pioglitazone,(±)-5[4[2(5-ethyl-2-pyridyl)ethoxy]benzyl]-thiazolidine-2,4-dione-monohydrochloride), purchased from Searle, Pvt, Ltd., Pakistan

2. Streptozotocin, (STZ) purchased from Nova Laboratories, Sdn, Bhd., Malaysia.

3. Vasoactive agents: NA (Sanofi Winthrop, Surrey, UK), PE (Knoll, Nottingham, UK), ME (Wellcome, London, UK) and Ang II (CIBA-GEIGY, Basel, Switzerland) were used in the

renal vasoconstrictor experiment. Noradrenaline acts on α1 and α2-adrenoceptors; whereas PE acts selectively to activate α1A-, α1B-, and α1D-adrenoceptor subtypes and ME activates only α1A adrenoceptors-subtype [23]. Angiotensin II (ANG II) produces vasoconstriction due to binding to AT1 receptors.

4. Full length recombinant adiponectin purchased from Chemtron Biotechnology Sdn, Bhd was dissolved in 200μl phosphate buffer saline.

## Experimental protocol and streptozotocin, pioglitazone and adiponectin pre-treatment

Type 1 diabetes was induced using a single intra-peritoneal injection (I/P) of (STZ) (Nova Laboratories, Sdn, Bhd, Malaysia), 40 mg/kg body weight, dissolved in citrate buffer (10 mM, pH 4.5) [24]. The injected rats were given glucose (5%) in drinking water for the first 48 hours after injection to offset the early hypoglycemic shock. Blood glucose levels were evaluated using a standard Glucometer (Free Style, Abbott, Malaysia) on day 7 of post-streptozotocin injection (between 9:00–9:30) and rats with glucose levels > 300 mg/dL at the 8th day were selected for the experiment. (Glucometer Free Style, Aabbott, Malaysia. Blood pressure of experimental rats was observed using the tail cuff method and rats with systolic blood pressure only higher than 150 mmHg were selected and randomized into groups and used for the study. The systolic blood pressure (SBP), diastolic blood pressure (DBP), mean arterial pressure (MAP), heart rate (HR) were measured non-invasively using the CODA equipment (Kent Scientific Corporation, Torrington, CT) on days 0, 8, 21 and 28. Overnight urine collections were performed using the metabolic cages before the start of treatment protocol on days 0, 21 and 28 of the experiment. Water intake and body weight were also measured on each day of urine collection. Blood samples (2ml) were collected from the tail vein on the same days and plasma was obtained following centrifugation. Plasma and urine sample were stored at -30˚C for biochemical analysis. Glomerular filtration rate (GFR) was calculated as creatinine clearance. Serum levels of adiponectin were measured on the acute surgery day [29] using an ELISA, (Chemtron Biotechnology Sdn, Bhd.) according to the manufacturer's protocol. To observe the effect of pioglitazone and adiponectin on altered vascular responsiveness in diabetic state *in-vivo*, the vasopressor responses to vasoactive agents were examined in all the experimental groups.

## Acute haemodynamics

Overnight fasted rats (water ad-libitum) were anaesthetized with 60 mg/kg intra-peritoneal (I. P). sodium pentobarbitone (Nembutal®, CEVA, Libourne, France). A mid-line incision was done for trachea exposure and cannulation using PP240 (Protex, Kent, UK). Carotid artery was cannulated (PE 50, Portex, Kent, UK) and coupled to a pressure transducer (P23 ID Gould, Statham Instruments, UK) connected to computerized data acquisition system (Power Lab®, AD Instruments, Sydney, Australia) for blood pressure measurement. Similarly left jugular vein was cannulated (PE 50, Portex, Kent, UK) for infusion of doses of anesthesia and vasoactive agents (Perfusor secura FT 50ml, B. Braun). A mid line incision was carried out for the exposure of left kidney. On the dorsal surface of posterior end of kidney, a laser Doppler flow probe (OxyFlow, AD Instruments) was placed for measurement of renal cortical blood perfusion (RCBP) continuously throughout the experiment which was directly linked to the data acquisition system A cannula (PE50, Portex) was inserted via the left common iliac artery, in a way for the close entry of its tip to access the renal artery for administration of vasoactive agents, such as noradrenaline (NA), phenylephrine (PE), methoxamine (ME) and angiotensin

II (Ang II) into the renal artery. The baseline measurement of renal arterial pressure was monitored through another pressure transducer (model P23 ID Gould; Statham Instruments) linked to a computerized data acquisition system (Power Lab; AD Instruments). Meanwhile 6ml/kg/hr saline was continuously infused for keeping the cannula patent. One hour stabilization period was spared before baseline systemic haemodynamics values were acquired, which comprised systolic blood pressure (SBP), diastolic blood pressure (DBP), mean arterial blood pressure (MAP), heart rate (HR) and renal cortical blood perfusion (RCBP). The baseline blood pressure values were determined from the values at the beginning of each response. Once baseline measurements were acquired over 30 min, dose–response curves were generated as follows:

NA at 25, 50 and 100 ng, PE at 0·25, 0·50 and 1 μg, ME at 0·5, 1 and 2 μg and Ang II at 2·5, 5 and 10 ng. A washout period of 10 min was allowed between each agonist.

The acute haemodynamics study and acute surgical technique was performed as our previous laboratory studies protocols [20, 21].

## Biochemical analysis of stored plasma and urine samples

Plasma and urinary creatinine concentrations were measured spectrophotometrically (Jaffe's reaction) while sodium and potassium concentrations were measured using a flame photometer (Jenway Ltd., Felsted, UK). Plasma levels of adiponectin were measured on the acute surgery day 29 using an ELISA kit, (Chemtron Biotechnology Sdn, Bhd) according to the manufacturer's protocol. Creatinine clearance (Cr.Cl), fractional excretion of sodium (FENa), and absolute urinary sodium excretion (UNa V) and NA: K ratio was calculated using the standard equations. Urinary sodium-to-potassium ratio was calculated by dividing the urinary sodium concentration by potassium concentration.

## Statistical analysis

The data obtained in this study is expressed as Mean ± SEM. The vasoconstrictor responses caused by adrenergic agonists and Ang II were taken as the average values caused by each dose of the agonists administered in ascending and descending orders. We operate two-way ANOVA followed by the Bonferroni post hoc test using the statistical package Superanova (Abacus Inc., Sunnyvale, CA, USA) for the statistical analysis of the dose–response data. Moreover, one-way ANOVA followed by the Bonferroni post hoc test analysis for baseline haemodynamics parameters measurement and body weight, fluid intake, urine flow rate (UFR), plasma adiponectin were analyzed using repeated measures for differences between treatment period weeks. Statistical difference of 5% was considered significant among the mean values.

## Results

### Physiological and biochemical indices

At the beginning of the treatment protocol the basal body weight was the same in all groups (Table 1), whereas, the age-dependent body weight gains during the treatment period were not altered between the vehicle treated WKY and SHR groups (all P>0.05). All STZ injected animals developed diabetes, whereas there was a significantly higher blood glucose level in STZ treated SHRs as shown in Table 1 (P<0.05). Similarly it was observed that SHR diabetic and SHR diabetic treated groups had higher levels of blood glucose on day 8, 21 and 28 as compared to SHR+STZ on day 28. No significant effect was observed on blood glucose levels in groups treated with pioglitazone, adiponectin and combination of adiponectin with pioglitazone during and at the termination of the experimental protocol (P>0.05). The body weight of

**Table 1. Effect of exogenous pioglitazone, adiponectin, and a combination of pioglitazone and adiponectin on body weight, water intake, urine flow rate (UFR) and blood glucose concentration in diabetic Spontaneously hypertensive rats.**

| Parameters | Groups | Days of Observation | | | |
|---|---|---|---|---|---|
| | | Day 0 | Day 8 | Day 21 | Day 28 |
| Body Weight (g) | WKY | 245±5 | 250±7^ | 275±4* | 289±8* |
| | SHR | 242±3 | 248±6^ | 267±8* | 284±9* |
| | SHR+STZ | 245±3 | 200±5*^ | 208±7*^ | 209±10* |
| | SHR+STZ+Pio | 254±6 | 205±4* | 207±7* | 217±5* ж |
| | SHR+STZ+Adp | 252±4 | 213±7* | 215±9* | 206±4* |
| | SHR+STZ+Adp+Pio | 247±5 | 201±9* | 200±10* | 209±5* |
| Water intake (ml/d) | WKY | 43±1 | 44±2 | 45±3 | 45±2 |
| | SHR | 32±2^ | 34±2! | 34±3! | 37±4! |
| | SHR+STZ | 33±2 | 48±3^* | 48±2^* | 59±3^* |
| | SHR+STZ+Pio | 34±2 | 48±3* | 50±2* | 50±3* |
| | SHR+STZ+Adp | 36±2 | 46±3* | 47±2* | 57±3* |
| | SHR+STZ+Adp+Pio | 34±2 | 45±3* | 49±2* | 55±3* |
| UFR (mL/min/100 g) | WKY | 3.84±0.44 | 3.63±0.21 | 3.92±0.21 | 3.95±0.21 |
| | SHR | 3.10± 0.05 ! | 2.98±0.09 ! | 2.94±0.54 ! | 2.93±0.43! |
| | SHR+STZ | 3.09±0.04 | 12.35±0.52^ | 12.79±0.62^ | 13.18±0.04^ |
| | SHR+STZ+Pio | 3.07±0.03 | 13.58±0.5* δ | 13.57±0.2* δ | 13.58±0.3* δ ж |
| | SHR+STZ+Adp | 3.06±0.02 | 13.28±0.49* δ | 13.35±0.15* δ | 16.25±0.13* δ |
| | SHR+STZ+Adp+Pio | 3.09±0.03 | 13.57±0.29* δ | 13.58±0.24* δ | 20.28±0.29* δ ζ |
| Blood glucose (mg/dl) | WKY | 89±3 | 88±2 | 86±2 | 88±3 |
| | SHR | 91±3 | 90±2 | 88±3 | 89±3 |
| | SHR+STZ | 90±3 | 460±18*^ | 471±14*^ | 489±25*^ |
| | SHR+STZ+Pio | 90±5 | 471±21* | 477±19* | 474±15* |
| | SHR+STZ+Adp | 88±2 | 465±19* | 462±18* | 484±27* |
| | SHR+STZ+Adp+Pio | 86±3 | 479±21* | 486±18* | 480±22* |

Notes: Physiological parameters data. The values are presented as mean±S.E.M. (n = 6) in each group and were analyzed by repeated measure one-way ANOVA followed by *Bonferroni post hoc* test. Values with (P<0.05) were statistically significant during and at the end of treatment.

^ P<0.05 of WKY and SHR control to SHR diabetic control group.

! P<0.05 of SHR and WKY control.

* P<0.05 versus day 0 of the respective group

ж P<0.05 versus diabetic Pio and Adp groups.

δ P<0.05 versus diabetic Pio, Adp, and Pio+Adp to SHR diabetic group.

ζ P<0.05 versus diabetic Adp group in comparison to diabetic Pio+Adp group at day 21 & 28.

STZ-diabetic rats treated with pioglitazone, adiponectin and combination of adiponectin with pioglitazone did not change significantly. (P>0.05), (Table 1). A modest increase in body weight of non-diabetic groups was observed, but was not statistically significant different between WKY & SHR control groups on all four days of observation (P>0.05). Apart from this observation, it was also noticed that the body weight of SHR control rats increased with time as the body weight on day 8, 21 and day 28 was significantly higher as compared to day 0. In contrast, the body weight of SHR diabetic group decreased on day 8, 21 & day 28 as compared to day 0 (P<0.05). Similarly SHR+STZ+Pio, SHR+STZ+Adp and SHR+STZ+Adp+Pio treated groups followed the same pattern of decreasing body weight irrespective of the treatment with adiponectin, pioglitazone and combination of adiponectin with pioglitazone on day 8, 21 and 28 as compared to day 0, (Table 1).

The mean values of daily water intake of all the eight groups are offered in (Table 1), and were observed on four occasions during the study period, i.e. on day 0, day 8, day 21, and day 28. There was no significant difference of fluid intake in WKY control group on all four days of observation (P>0.05). It was also noticed that water intake was significantly lower in the SHR control group in comparison with the WKY control group on all 4 days of observation (P<0.05). Moreover, with the onset of diabetes the SHR diabetic rats showed higher water intake on day 8, 21 & 28 as compared to day 0. Similarly SHR+STZ+Pio, SHR+STZ+Adp and SHR+STZ+Adp+Pio treated groups exhibited with polydipsia as compared to SHR control group on day 8, 21 and 28 in comparison to day 0 (P<0.05). No significant effect was observed on water intake in groups treated with pioglitazone, adiponectin and combination of adiponectin with pioglitazone during and at the end of the experiment (P>0.05), (Table 1).

Urine flow rate was significantly lower in the SHR control compared to the WKY control group on day 0, 8, 21 and 28. (P<0.05), (Table 1). Contrary to SHR control group readings, it was observed that induction of diabetes with STZ, SHR+STZ treated rats showed polyuria as compared to SHR control on day 8, 21 and day 28 (P<0.05). It was observe that SHR+STZ +Pio and SHR+STZ+Adp treated groups show significant difference on day 21 and 28 of the experiment as compared to SHR+STZ group (P<0.05). Moreover, significant increase in urine output was observed in and SHR+STZ+Adp+Pio treated groups on day 28 only as compared to SHR+STZ, SHR+STZ+ADP and SHR+STZ+Pio treated groups (P<0.05), (Table 1).

## Systemic haemodynamics

We measured the variations in the systolic blood pressure (SBP), diastolic blood pressure (DBP), mean arterial pressure (MAP and heart rate (HR) were monitored non-invasively for the conscious rats on day 0, 21 and 28 of the experiment (Table 2). SBP, DBP and MAP and HR on all days of observations were significantly higher in SHR control as compared to WKY control group (all P<0.05).

Moreover, the diabetic model of SHRs expressed higher SBP and MAP as compared to SHR control groups on day 8, 21 and 28, (P<0.05). Increase in SBP and MAP in diabetic SHR was blunted by the treatment of 10mg/kg/day of pioglitazone for 28 days and 2.5μg/kg/day of adiponectin for 7 days. The SHR+STZ+Pio and SHR+STZ+Adp groups exhibited decreased SBP on day 28 as compared to SHR+STZ group (P<0.05) and comparable to WKY control group (Table 2). Whereas, the combined treatment of diabetic SHRs in SHR+STZ+Adp+Pio significantly resulted in further lowering significantly decreased SBP, DBP and MAP on day 28 as compared to separate treatment in SHR+STZ+Adp and SHR+STZ+Pio. (all P<0.05), (Table 2). Similarly, the heart rate of SHR control groups of rats remained significantly higher as compared to WKY control throughout the study period (all P<0.05). The heart rate of diabetic SHR control group was higher as compared to SHR control on day 21 and 28 (P<0.05). Treating diabetic SHR with pioglitazone or adiponectin significantly decreased heart rate in SHR+STZ+Pio and SHR+STZ+Adp groups as compared to SHR+STZ group on day 28 only, (P<0.05). However, the intensity to reduce the heart rate in SHR+STZ+Adp was significantly more as compared to SHR+STZ+Pio group. No significant effect was observed in case of combined treatment of adiponectin with pioglitazone in SHR+STZ+Adp+Pio group (P>0.05), (Table 2).

**Renal cortical blood perfusion.** The renal cortical blood perfusion (RCBP) was measured on day 29 of the acute study. It was observed that SHR diabetic group exhibited lower RCBP as compared to WKY and SHR control group (133±12 vs. 247 ±11, 167±9) bpu respectively (P<0.05). In SHR diabetic groups treated with pioglitazone, adiponectin separate treatment, RCBP was significantly higher as compared to SHR diabetic control group, i.e. SHR+STZ+Pio,

**Table 2. Non-invasive blood pressure measurements of effect of exogenous pioglitazone, adiponectin, and a combination of pioglitazone and adiponectin on systolic, diastolic, mean arterial pressure and heart rate in diabetic Spontaneously hypertensive rats.**

| Parameters | Groups | Days of Observation | | | |
|---|---|---|---|---|---|
| | | Day 0 | Day 8 | Day 21 | Day 28 |
| Systolic blood pressure (mmHg) | WKY | 118±5 | 117±2 | 117±2 | 120±3 |
| | SHR | 159±4 ! | 164±6 ! | 162±3 ! | 157±8 ! |
| | SHR+STZ | 161±5 | 173±3^* | 177±4^* | 175±3^* |
| | SHR+STZ+Pio | 165±6 | 179±4* | 155±3* δ ж | 148±6* δ ж |
| | SHR+STZ+Adp | 162±4 | 175±2* | 174±3*# | 138±4* δ |
| | SHR+STZ+Adp+Pio | 163±5 | 177±4* | 154±4* δ | 134±3*δ ζ |
| Diastolic blood pressure(mmHg) | WKY | 79±3 | 84±4 | 80±3 | 86±5 |
| | SHR | 119±6 ! | 117±7 ! | 108±7 ! | 120±6 ! |
| | SHR+STZ | 117±3 | 119±2 | 120±3 | 118±4 |
| | SHR+STZ+Pio | 116±3 | 119±3 | 109±5* δ | 106±4* |
| | SHR+STZ+Adp | 117±4 | 120±4 | 118±4 | 96±2* δ# ж |
| | SHR+STZ+Adp+Pio | 118±3 | 120±2 | 111±3* δ | 98±2*δζ |
| Mean arterial pressure (mmHg) | WKY | 92±8 | 97±4 | 92±4 | 97±5 |
| | SHR | 132±5 ! | 133±4 ! | 126±5 ! | 132±6 ! |
| | SHR+STZ | 132±7 | 137±5* | 143±4^* | 144±5^* |
| | SHR+STZ+Pio | 132±4 | 139±4* | 124±4* δ | 120±5* δδ ж |
| | SHR+STZ+Adp | 132±3 | 138±7* | 137±6* | 110±8* δ |
| | SHR+STZ+Adp+Pio | 133±5 | 139±5* | 125±3* δ | 105±6* δ ζ |
| HR (beat/min | WKY | 312±10 | 310±4 | 309±11 | 303±8 |
| | SHR | 386±9 ! | 390±10 ! | 392±14 ! | 389±11 ! |
| | SHR+STZ | 387±5 ! | 394±7 | 402±4^* | 407±5^* |
| | SHR+STZ+Pio | 388±6 | 400±3* | 380±3* δ ж | 367±4* δ ж |
| | SHR+STZ+Adp | 383±3 | 398±5* | 395±4* | 356±3*δ |
| | SHR+STZ+Adp+Pio | 387±7 | 403±5* | 377±6*δ | 351±5*δ |

Notes: Conscious blood pressure measurement data. The values are presented as mean±S.E.M. (n = 6) in each group and were analyzed by repeated measure one-way ANOVA followed by *Bonferroni post hoc* test. Values with (P<0.05) were considered statistically significant during and at the end of treatment.

^ P<0.05 of WKY and SHR control to SHR diabetic control group.

! P<0.05 of SHR and WKY control.

* P<0.05 versus day 0 of the respective group.

ж P<0.05 versus diabetic Pio and Adp groups.

δ P<0.05 versus diabetic Pio, Adp, and Pio+Adp to SHR diabetic group.

ζ P<0.05 versus diabetic Adp group to diabetic Pio+Adp group at day 21 and 28.

SHR+STZ+Adp vs. SHR+STZ: 166±12, 187±9 vs. 133±12) bpu (P<0.05). Moreover, the RCBP in SHR+STZ+Adp group was significantly higher as compared to SHR+STZ+Pio group, but still remained significantly lower as compared to WKY control group. The combination therapy of adiponectin further increase the RCBP in SHR+STZ+Adp+Pio group (209±12)bpu, but this increase in RCBP was significantly higher as compared to values found in SHR+STZ+Adp and SHR+STZ+Pio groups (P<0.05), (Fig 1).

## Plasma adiponectin measurement

Plasma adiponectin concentration were measured on day 28 only in SHR and SHR diabetic treated groups. A significant increases of plasma adiponectin concentration was observed in SHR and WKY control groups as compared to SHR+STZ group, (P<0.05). Moreover, diabetic

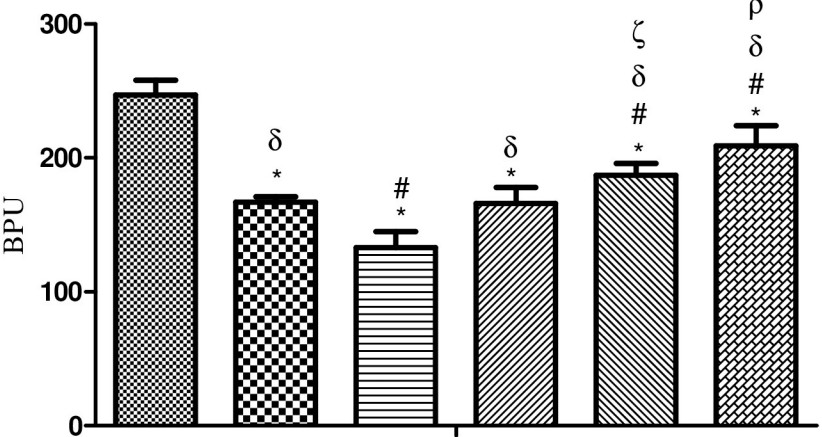

Baseline renal cortical blood perfusion

WKY        SHR        SHR+STZ        SHR+STZ+PIO        SHR+STZ+ADP

SHR+STZ+PIO+ADP

**Fig 1. Baseline renal cortical blood perfusion of WKY, SHR, SHR diabetic control and SHR diabetic treated rats.** The values are presented as mean±S.E.M. (n = 6) in each group and were analyzed by one-way ANOVA followed by *Bonferroni post hoc* test. Values with (P<0.05) were considered statistically significant.

SHRs treated with pioglitazone (10mg/kg/day) for three weeks significantly increased the plasma adiponectin levels in the SHR+STZ+Pio as compared to the control SHR+STZ group (p<0.05) but the magnitude of increase was significantly (both P<0.05) lower compared to one week adiponectin treatment at a dose rate of 2.5µg/kg/day. However, the extent of increase in plasma concentration of adiponectin in combined treatment of SHR+STZ+Adp+Pio group was significantly more as compared to their separate treatment groups (P>0.05), (Fig 2).

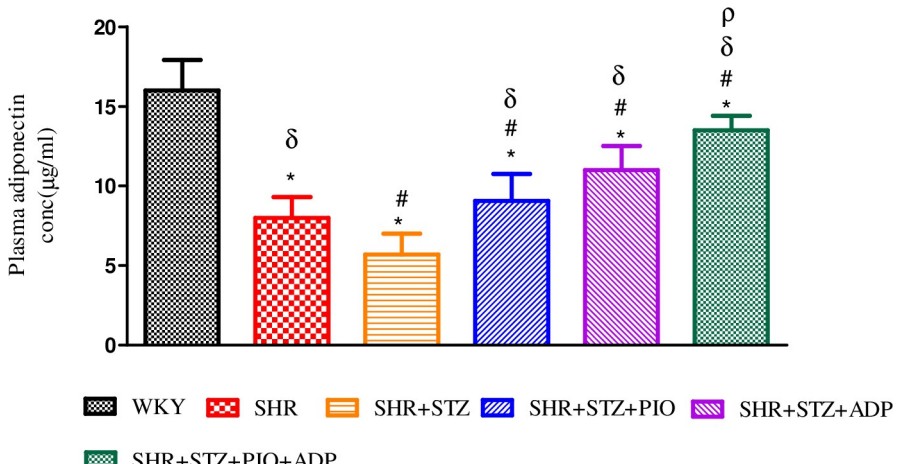

WKY    SHR    SHR+STZ    SHR+STZ+PIO    SHR+STZ+ADP

SHR+STZ+PIO+ADP

**Fig 2. Adiponectin plasma concentration of WKY, SHR, SHR diabetic control and SHR diabetic treated rats, expressed as µg/ml values.** The values are presented as mean±S.E.M. (n = 6) in each group and were analyzed by one-way ANOVA followed by *Bonferroni post hoc* test. Values with (P<0.05) were considered statistically significant.

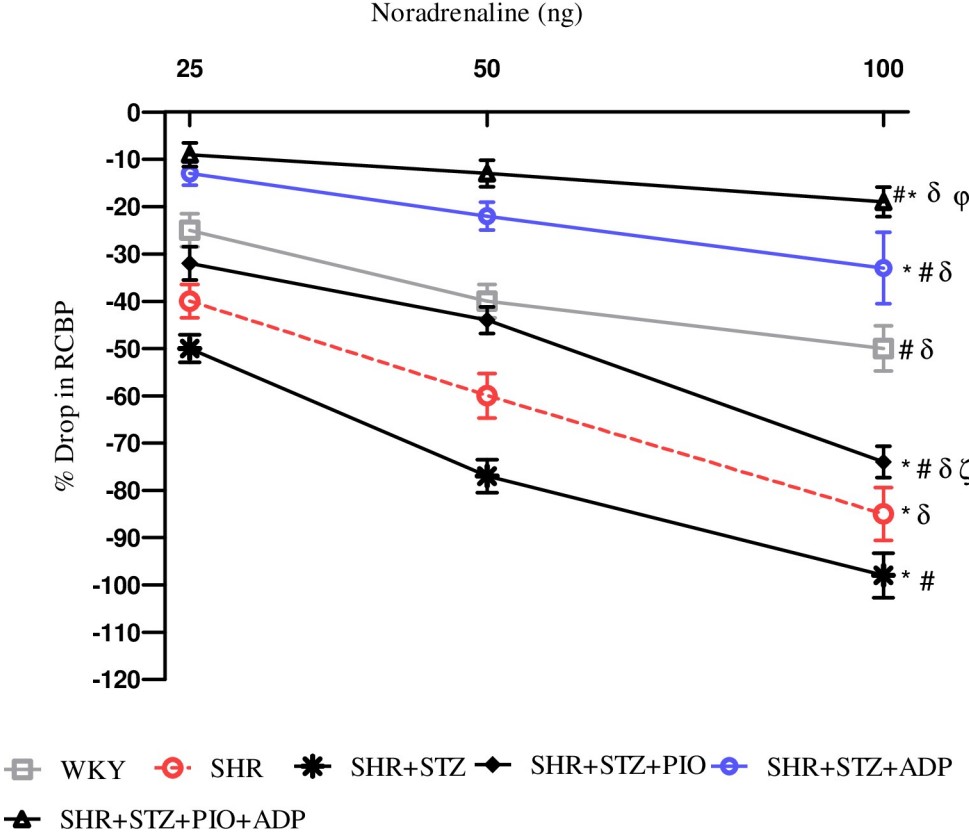

**Fig 3. Dose–response curve of the renal vasoconstrictor responses (RCBP % drop) to graded doses of NA in WKY, SHR, SHR diabetic control and SHR diabetic treated rats.** The values are presented as mean±S.E.M. (n = 6) in each group and were analyzed by two-way ANOVA followed by *Bonferroni post hoc* test. Values with (P<0.05) were considered statistically significant.

### Acute renal vasoconstriction responses

During the acute experiment, the responses of the renal vasculature were observed and owed to the vasoactive effect of the agonists administered. The intravenous injection of saline in the same volume with the one used for the agonist dose had no effect on the systemic and renal blood pressure. Dose dependent renal cortical vasoconstriction were assessed after intra-renal injection of NA, PE and ME (Figs 3–5), in all experimental groups of study.

The overall mean % decrease in RCBP was significantly larger in the SHR+STZ rats compared to the SHR and WKY control rats to intra-renal administration of NA, PE and ME (all P<0.05), (Figs 3–5). Treatment with pioglitazone in SHR+STZ+Pio had a significantly lesser decrease in RCBP to NA (Fig 3), PE, (Fig 4) (P<0.05), but not to ME, (Fig 5), (P>0.05). However, one week treatment with adiponectin alone and in combination with pioglitazone in diabetic SHR significantly reduced magnitude of adrenergically induced reductions in RCBP as compared to untreated diabetic SHR after intra-renal administration of NA (Fig 3), PE (Fig 4) and ME, (Fig 5), (all P<0.05). No significant difference was observed in SHR+STZ+Pio+Adp as compared to SHR+STZ+Adp groups as far as overall percentage drop was concerned after intra-renal administration of adrenergic agonists (P>0.05). The values obtained in the adiponectin treated groups only were comparable to WKY control groups.

**Angiotensin II (Ang II).** The dose dependent and overall Ang II-induced renal vasoconstriction in control WKY was significantly smaller (P<0.05) as compared to SHR control and

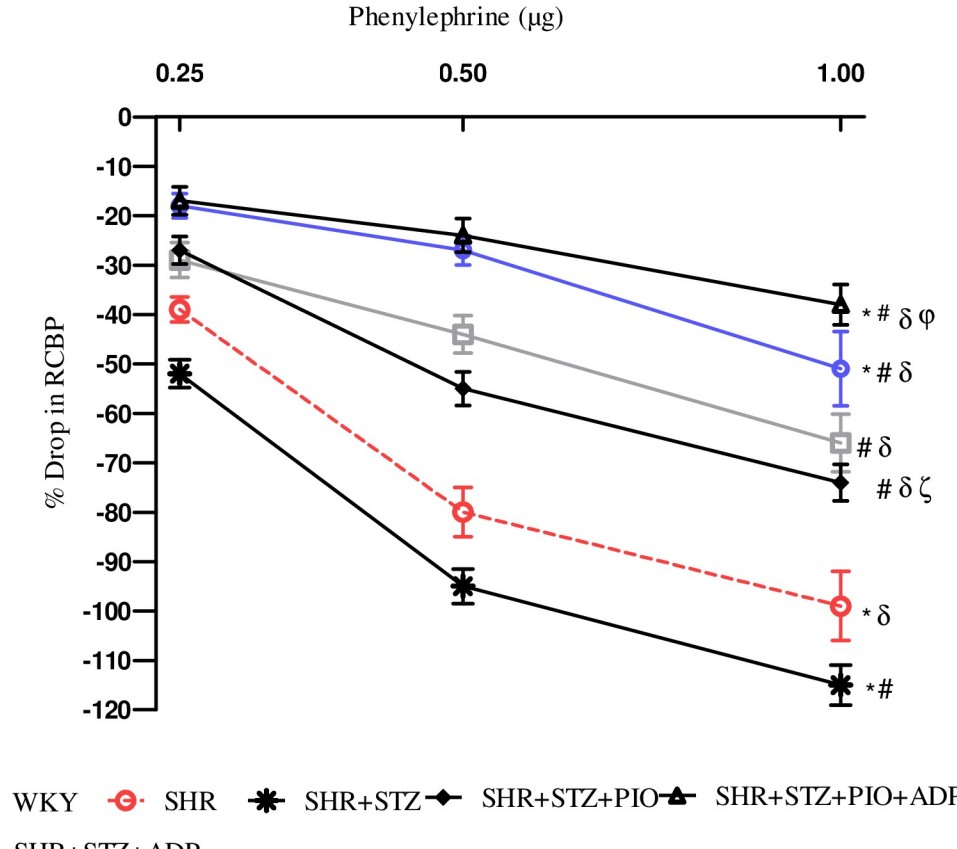

**Fig 4. Dose–response curve of the renal vasoconstrictor responses (RCBP % drop) to graded doses of PE in WKY, SHR, SHR diabetic control and SHR diabetic treated rats.** The values are presented as mean±S.E.M. (n = 6) in each group and were analyzed by two-way ANOVA followed by *Bonferroni post hoc* test. Values with (P<0.05) were considered statistically significant.

SHR diabetic control (Fig 6). Moreover, the extent of Ang II induced decrease in RCBP was significantly blunted (P<0.05) in the adiponectin treated groups as compared to SHRs. The renal vasoconstriction induced by Ang II in the pioglitazone treated group was significantly higher (P<0.05) comparatively to combination treatment of pioglitazone with adiponectin and compared to the SHR+CNT group (P<0.05), (Fig 6).

## Renal haemodynamics parameters

Renal haemodynamics parameters observations were made on day 0, 8,21 and 28 of the study. The absolute urinary sodium excretion ($U_{Na}V$) and basal creatinine clearance (Cr.Cl) of the SHR+CNT group were significantly lower versus WKY control group on the experimental days all days of observation. (Fig 2A), (P<0.05), whereas SHR+STZ group exhibited higher Cr. Cl and $U_{Na}V$ on day 8, 21 and 28 as compare to SHR+CNT (P<0.05). However, SHR+STZ +Pio, caused absolute sodium excretion to increase in comparison of SHR+STZ on day 21. Moreover, SHR+STZ+Adp showed greater increase in absolute sodium excretion on day 28 as compared with SHR+STZ group SHR+STZ+Adp+Pio groups showed higher levels of creatinine clearance were observed when compared to SHR+STZ group on day 28 only. Adiponectin with pioglitazone in SHR+STZ+Adp+Pio groups exhibited higher creatinine clearance as compared to the separate treatments. (P<0.05), (Table 3). The fractional sodium excretion

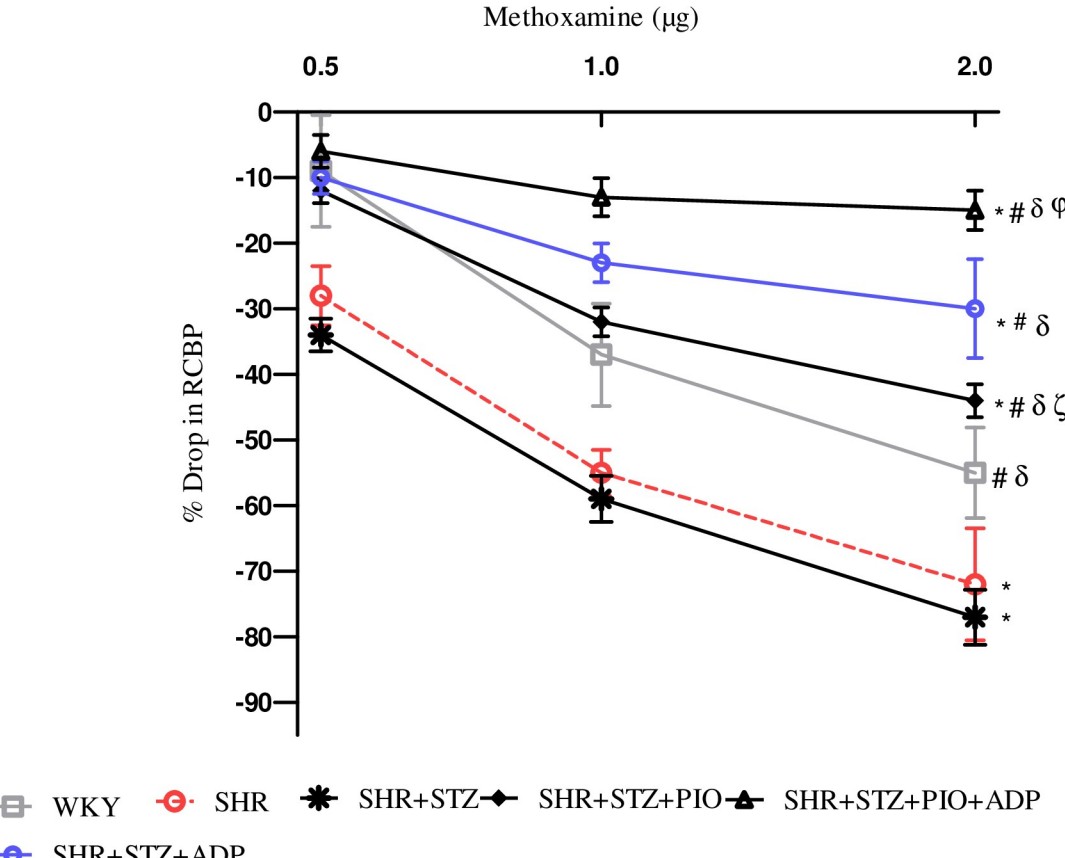

**Fig 5. Dose–response curve of the renal vasoconstrictor responses (RCBP % drop) to graded doses of ME in WKY, SHR, SHR diabetic control and SHR diabetic treated rats.** The values are presented as mean±S.E.M. (n = 6) in each group and were analyzed by two-way ANOVA followed by *Bonferroni post hoc* test. Values with (P<0.05) were considered statistically significant.

($FE_{Na}$) of all the SHR groups under observation were significantly lowers on days 0, 8, 21 and 28 versus WKY control group (P<0.05). Fractional sodium excretion of SHR+STZ group was higher as compared to SHR control group on day 8, 21 and 28 (P<0.05). However, it was observed that SHR+STZ+Pio groups caused fractional sodium excretion ($FE_{Na}$) to increase as compared to SHR+STZ on day 21, and 28 (P<0.05). Moreover, SHR+STZ+Adp group showed greater increase in fractional sodium excretion on day 28 as compared to SHR+STZ group (P<0.05). However, values in SHR+STZ+Adp+Pio group were significantly higher as compared to SHR+STZ+Adp groups on day 28 (P<0.05), (Table 3).

**Urinary potassium and sodium to potassium ratio.** Observations for changes in urinary potassium were made on four occasions *i.e.* day 0,8,21 and 28. The urinary potassium concentration of all SHR group was significantly higher as compared to WKY control group (P<0.05). However, SHR+STZ group exhibited higher urine potassium concentration as compared to SHR+CNT group on day 8, 21 and 28 of the study. As study progressed, SHR+STZ +Pio group significantly decreased urinary potassium concentration on day 21 and 28, whereas, SHR+STZ+Adp expressed greater decrease in urinary potassium concentration on day 28 only versus SHR+STZ group (P<0.05). However, it was observed that urinary potassium values significantly reduced in SHR+STZ+Pio+Adp as compared to SHR+STZ+Pio and SHR+STZ+Adp groups on day 28 (P<0.05), (Table 3).

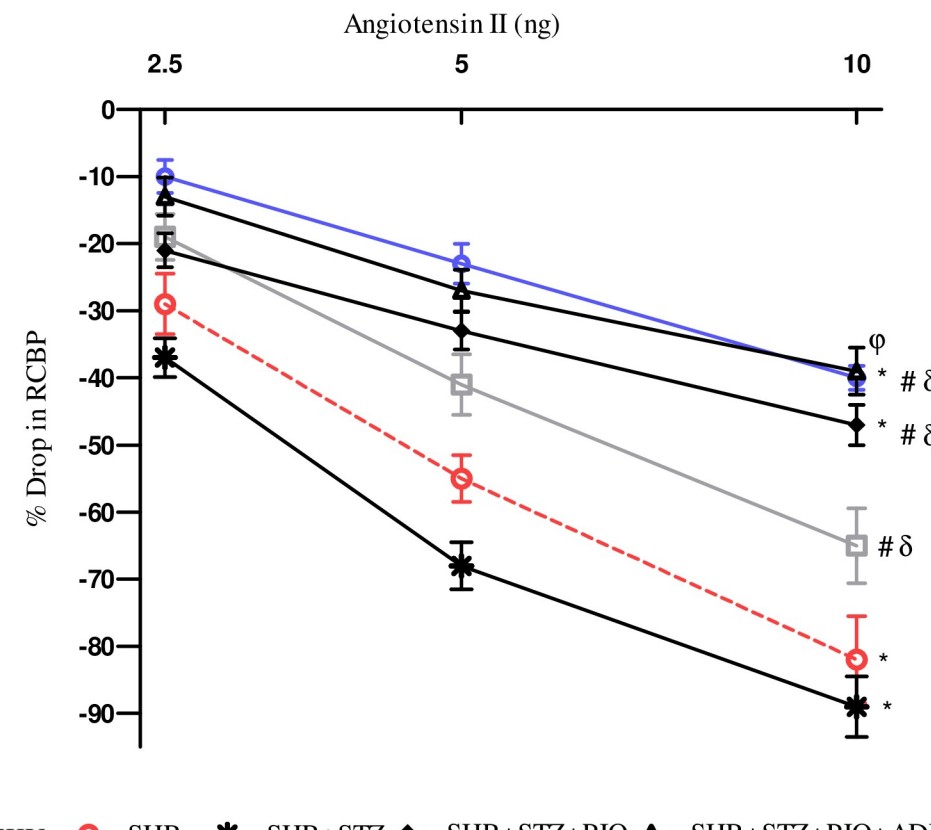

**Fig 6. Dose–response curve of the renal vasoconstrictor responses (RCBP % drop) to graded doses of ANG II in WKY, SHR, SHR diabetic control and SHR diabetic treated rats.** The values are presented as mean±S.E.M. (n = 6) in each group and were analyzed by two-way ANOVA followed by *Bonferroni post hoc* test. Values with (P<0.05) were considered statistically significant.

Similarly changes in urinary sodium to potassium ratio were measured on four points during the study. No significant difference was observed among all SHR groups on day 0, 8, 21, 28 (P>0.05), but found to be lower in values versus WKY control group on all four days of observation (P<0.05). A significant decrease of urinary sodium potassium ratio was observed in SHR+STZ as compared to SHR+CNT group on day 8, 21 & 28 (P>0.05), (Table 3). Moreover, it was observed, that in SHR+STZ+Pio groups, Na:K ratio increased as compared to SHR+STZ on day 21 and 28 (P<0.05). Moreover, SHR+STZ+Adp showed greater increase in Na:K ratio on day 28 as compared to SHR+STZ group (P<0.05). In addition, it was noticed that values in SHR+STZ+Adp+Pio group were significantly higher as compared to SHR+STZ+Adp groups on day 28 of observation (P<0.05), (Table 3).

## Discussion

In this study we sought to determine that whether in a diabetic hypertensive rat model, exogenous treatment of adiponectin and pioglitazone (full PPAR-γ agonist) separately and in combination would reduce B.P or regulate renal haemodynamics receptiveness to vasoconstrictor agents to a similar degree or larger extent following combined treatments. We observed numerous significant findings in this experiment. Primarily, urinary excretion and plasma

**Table 3. Effect of exogenous pioglitazone, adiponectin, and a combination of pioglitazone and adiponectin on creatinine clearance (Cr.Cl), urinary creatinine (mg/dl), urinary sodium concentration (urinary Na), absolute urinary sodium excretion ($U_{Na}V$) fractional sodium excretion(FENa), urinary potassium (urinary K) and urinary sodium to potassium ratio (Na:K) in diabetic Spontaneously hypertensive rats.**

| Parameters | Groups | Days of Observation | | | |
|---|---|---|---|---|---|
| | | Day 0 | Day 8 | Day 21 | Day 28 |
| Cr.Cl(ml/min/kg) | WKY | 0.45±0.01 | 0.41±0.02 | 0.42±0.00 | 0.41±0.02 |
| | SHR | 0.30±0.02 ! | 0.33±0.01 ! | 0.32±0.01 ! | 0.33±0.01 ! |
| | SHR+STZ | 0.32±0.01 | 0.90±0.05^* | 0.92±0.07^* | 1.15±0.11^* |
| | SHR+STZ+Pio | 0.31±0.02 | 0.94±0.47* | 0.95±0.05* | 1.45±0.02* δ |
| | SHR+STZ+Adp | 0.30±0.05 | 0.90±0.04* | 0.09±0.08* | 1.87±0.14* δ |
| | SHR+STZ+Adp+Pio | 0.31±0.02 | 0.93±0.05* | 0.96±0.57* | 1.94±0.99* δ |
| Urinary Cr (mg/dl) | WKY | 151.87±1.88 | 152.85±2.89 | 156.23±1.02 | 157.05±1.21 |
| | SHR | 99.34±1.57 ! | 103.57±2.69 ! | 104.23±2.19 ! | 102.39±2.58 ! |
| | SHR+STZ | 100.38±1.59 | 77.81±3.51 ^* | 79.25±3.67 ^* | 81.74±2.69 ^* |
| | SHR+STZ+Pio | 102.38±2.54 | 79.36±2.58* | 81.58±1.93* | 86.23±3.13* δ ж |
| | SHR+STZ+Adp | 101.87±2.71 | 77.39±2.67* | 79.58±2.19* | 93.57±1.67* δ |
| | SHR+STZ+Adp+Pio | 103.67±2.57 | 80.23±2.49* | 80.31±1.97* | 104.58±2.41 δ ζ |
| Urinary Na conc. (mmol/L) | WKY | 137.84±1.34 | 139.23±1.55 | 140.25±2.35 | 141.25±2.89 |
| | SHR | 91.03±2.61 ! | 94.34±2.17 ! | 98.63±5.31 ! | 95.67±3.59 ! |
| | SHR+STZ | 95.61±2.53 | 117.89±2.57^* | 121.31±2.67^* | 120.97±2.53^* |
| | SHR+STZ+Pio | 95.37±2.47 | 117.37±3.37* | 120.37±3.57* | 123.61±2.37* ж |
| | SHR+STZ+Adp | 96.31±2.87 | 119.31±2.47* | 117.37±4.31* | 131.89±2.71* δ |
| | SHR+STZ+Adp+Pio | 96.31±1.98 | 115.28±2.57* | 123.25±2.43* | 142.37±3.15* δ ζ |
| $U_{Na}V$(mmol/hr/100g) | WKY | 0.22±0.01 | 0.23±0.01 | 0.23±0.01 | 0.24±0.00 |
| | SHR | 0.01±0.05 ! | 0.01±0.04 ! | 0.01±0.03 ! | 0.14±0.05 ! |
| | SHR+STZ | 0.01±0.03 | 0.06±0.03^* | 0.08±0.02^* | 0.10±0.04^* |
| | SHR+STZ+Pio | 0.01±0.03 | 0.07±0.04* | 0.11±0.02* δ | 0.14±0.07 δ ж |
| | SHR+STZ+Adp | 0.01±0.01 | 0.06±0.02* | 0.07±0.03* | 0.17±0.06 δ |
| | SHR+STZ+Adp+Pio | 0.01±0.04 | 0.07±0.03* | 0.11±0.06* δ | 0.23±0.05 δ ζ |
| FENa ((%) | WKY | 0.50±0.02 | 0.53±0.01 | 0.59±0.01 | 0.54±0.03 |
| | SHR | 0.35±0.03 ! | 0.37±0.03 ! | 0.37±0.02 ! | 0.34±0.01 ! |
| | SHR+STZ | 0.33±0.02 | 0.79±0.03^* | 0.99±0.05^* | 1.05±0.04^* |
| | SHR+STZ+Pio | 0.33±0.04 | 0.99±0.05* | 1.10±0.03* δ | 1.11±0.05* δ ж |
| | SHR+STZ+Adp | 0.32±0.04 | 0.84±0.04* | 0.86±0.02* | 1.12±0.03* δ |
| | SHR+STZ+Adp+Pio | 0.33±0.05 | 0.10±0.04* | 1.11±0.25* δ | 1.23±0.06* δ ζ |
| Urinary K (mmol/L) | WKY | 44.88±1.00 | 45.82±1.98 | 43.52±1.83 | 44.05±1.27 |
| | SHR | 55.61±1.82 ! | 55.17±1.47 ! | 56.19±1.77 ! | 56.97±1.33 ! |
| | SHR+STZ | 55.89±1.41 | 166.41±5.12^* | 191.23±4.14^* | 213.21±6.37^* |
| | SHR+STZ+Pio | 56.17±1.37 | 180.23±5.27* | 150.22±3.61* δ ж | 109.27±4.83* δ ж |
| | SHR+STZ+Adp | 55.18±1.28 | 184.49±3.69* | 199.41±5.19* | 115.27±5.87* δ |
| | SHR+STZ+Adp+Pio | 56.13±1.17 | 190.3±3.81* | 146.27±3.40* δ | 94.37±3.19* δ ζ |
| Na:K | WKY | 3.00±0.08 | 3.09±0.07 | 3.02±0.04 | 3.04±0.13 |
| | SHR | 1.35±0.14 ! | 1.37±0.21 ! | 1.32±0.18 ! | 1.41±0.32 ! |
| | SHR+STZ | 1.39±0.23 | 0.36±0.15^* | 0.32±0.25^* | 0.24±0.35^* |
| | SHR+STZ+Pio | 1.41±0.25 | 0.35±0.18* | 0.66±0.26*δ ж | 0.83±0.31* δ ж |
| | SHR+STZ+Adp | 1.39±0.25 | 0.35±0.19* | 0.32±0.16* | 1.29±0.12* δ |

(*Continued*)

**Table 3.** (Continued)

| Parameters | Groups | Days of Observation | | | |
|---|---|---|---|---|---|
| | SHR+STZ+Adp+Pio | 1.38±0.19 | 0.36±0.25* | 0.70±0.33* δ | 1.61±0.31 δ ζ |

Notes: Renal haemodynamics parameters data. The values are presented as mean±S.E.M. (n = 6) in each group and were analyzed by repeated measure one-way ANOVA followed by *Bonferroni post hoc* test. Values with (P<0.05) were statistically significant during and at the end of treatment.

^ P<0.05 of WKY and SHR control to SHR diabetic control group.

! P<0.05 of SHR and WKY control.

* P<0.05 versus day 0 of the respective group, ж P<0.05 versus diabetic Pio and Adp groups.

δ P<0.05 versus diabetic Pio, Adp, and Pio+Adp to SHR diabetic group.

ж P<0.05 versus diabetic Pio and Adp groups.

ζ P<0.05 versus diabetic Adp group in comparison to diabetic Pio+Adp group at day 21 & 28.

adiponectin concentrations were low in the diabetic SHRs, whereas, restored partially to those found in the WKY rats after pioglitazone administration which supports the previous findings, that through activation of PPAR-γ, adiponectin, prevents activation of the adrenoceptors and AT1 receptor [25]. We administered adiponectin at a rate sufficient to increase its plasma concentration and urinary excretion as compared to normotensive rats. Furthermore, the combined therapy of pioglitazone and adiponectin reduced B.P and improved renal haemodynamics and fluid excretion to a larger greater degree versus separate treatments. Our hypothesis is supported with these observations and further aims that both drugs prevent the ANGII actions along the signaling pathway. In addition, the renal vasoconstrictor responses to the ANGII and adrenergic agonists were increased in the diabetic SHRs as compared to WKY, whereas, these responses were reduced to a larger extent after adiponectin and pioglitazone treatments, which indicates that PPAR-γ and adiponectin receptors (adipo R1 and R2) interacts to depress the signaling following activation of adrenoceptors within the kidney vasculature.

In the present study, a single intra-peritoneal injection of streptozotocin (STZ) at a dose of 40 mg/kg was able to produce and maintain hyperglycemia in SHRs, since this genetic hypertensive model more susceptible to the diabetogenic effect of STZ than normal WKY [26]. Diabetes induced by high-dose STZ is similar to human type 1 diabetic model [27]. We observed that plasma adiponectin levels were reduced after induction of diabetes. Adiponectin is an insulin-sensitizing agent [28], whereas the treatment of SHRs with pioglitazone for 3 weeks significantly increased adiponectin levels with more values in pioglitazone and adiponectin combined treatment. This finding is parallel with the results of Thule [29], mentioning that adiponectin plasma concentration decreases in STZ- induced experimental rats. Moreover, in disparity to our report, it has been observed that plasma adiponectin concentration enhances in type 1 diabetes [30], which probably could be due to species differences and related to immune system, which if defective could lead to β-cell damage in STZ-induced diabetic SHRs [31]. We understand that pioglitazone, involves an adiponectin-dependent pathway and increase adiponectin levels, ameliorating insulin resistance and decreasing gluconeogenesis in the liver [6]. Thus, the glycaemic status remains unaffected in the present study by exogenously administered adiponectin and pioglitazone in separate and combined fashion.

In the experimentally induced diabetic rats, a significant reduction in the body weight was observed, whereas, STZ lead to same degree of hyperglycemia in all experimental groups. This finding is consistent with the earlier studies of our laboratory that the weight loss is one of the common happening in STZ induced diabetes in rats [32, 33]. In a state of experimental diabetes such as type 1, most of β -cells are destroyed leading to hypoinsulinemia and

hyperglycaemia due to the decreased utilization of the glucose by the tissues and will lead to reduction in body weight [34]. Polyuria and polydipsia are the other common phenomenon of diabetic state. Our lab previously also observed the same pattern of polyuria and polydipsia as we observed in this experiment [23]. An increased urine production caused by hyperglycemia, glycosuria and defective reabsorption of glucose by proximal renal tubules. Subsequently loss in plasma volume is replaced by intracellular water that leads to dehydration and increased thirst that causes polydipsia [35]. In our study, no significant effect was observed in weight gain and water intake of diabetic SHRs following all types of treatments. However, urine output was enhanced in diabetic animals after separate and combined treatment of adiponectin with pioglitazone. Adiponectin administration caused natriuresis, due to an inhibitory effect on anti-diuretic hormone release, which allows water to follow passively along sodium in diabetic SHRs [36].

In this study we also observed that SBP and MAP of diabetic SHRs was considerably higher as compared to non-diabetic animals throughout the study, whereas, following three weeks pioglitazone, one-week adiponectin administration alone and in combination with pioglitazone caused SBP and MAP to be significantly lower in diabetic SHRs. Our findings for this parameter is in parallel to those found in previous observations using pioglitazone in STZ-induced diabetic SD rats and fructose overloaded SD rats [37, 38]. It is evident that PPAR-γ agonists, including pioglitazone, are known to counteract the characteristics of diabetes and hypertension, activating via production of adiponectin which is mediator moderating oxidative stress through activation of signaling cascades, such as cAMP–PKA and AMPK–eNOS component [39] and reduction in sensitivity to Ang II [11] respectively. However, the extent of reduction of SBP & MAP in pioglitazone treated groups was lower as compared to adiponectin treated groups, which possibly could be due to enhanced production of nitric oxide (NO), stimulated by adiponectin [7], whereas, regulation of MAP and vascular tone depends upon nitric oxide, which acts as endothelium-derived molecule [40]. The secretion of renin from kidneys mediates production of angiotensin II, which inhibits the adiponectin production, thus offsetting the dilator action of NO reflecting an important interaction with RAAS in regulating B.P through NO production [12], and thus impacts on B.P control.

Interestingly, with the induction of diabetes, SHR diabetic groups exhibited decreased renal cortical blood perfusion (RCBP) and it is in agreement with our previous lab findings [33], which is due to the increased renovascular response to adrenergic and neural stimuli and activation of intra-renal RAAS [41]. The hyperglycemia induced by STZ lead to activation of local ANG II [42], which cause increased renovascular resistance, thus leads to reduced RCBP in diabetic rats.

In the present study, diabetic SHRs showed decreased values for reno-protective markers including urinary creatinine (urinary Cr) and clearance of creatinine (Cr.Cl), whereas, an increased urinary sodium (urinary Na conc.), urinary potassium (urinary K conc.), fractional excretion of sodium (FENa), which probably depicts the renal defects [43]. Diabetic rats expressed low urinary sodium (UNaV), demonstrating renal impairment of sodium handling which is in consent with the previous report of Khan [33]. We understand that kidneys regulates the long-term B.P through altered renal sodium handling to conserve the sodium-retaining characteristics [44], whereas, diabetes leads to renal pathologies via renal imbalance of vital electrolytes in the form of sodium retention [45]. Moreover, Na+/K-ATPase could be a factor for enhanced fractional sodium excretion [46]. We observed that UNaV was lower in the diabetic SHRs, which appeared to be an effect independent of higher B.P in SHRs [36], indicating long-term fluid retention and fluid volume. Meanwhile PPAR-γ is chiefly expressed in the collecting ducts and an important objective for TZDs [47], thus its stimulation could modify sodium resorption through activation of Na+/K+-ATPase system and epithelial sodium

channels [48], whereas, adiponectin effects through AMPK-activated pathway expressed in the kidney and this pathway appears to be necessary for the maintenance of normal renal physiology [49], and renal function. Adiponectin administration to the diabetic SHRs increased renal haemodynamics and fluid excretion similar to those achieved with pioglitazone. Firstly, the natriuresis (urinary sodium excretion) produced by adiponectin caused an increase in sodium and water excretion and could be contributory factor in reducing B.P. Secondly, adiponectin may inhibit Na+-K+-ATPase activity in renal proximal tubule cells of SHR diabetic rats, which supports our hypothesis that these drugs particularly adiponectin prevent the ANG II actions at various points along signaling pathway and in parallel with our previous findings [25].

Interestingly, increase in urinary Cr, Cr.Cl, FENa, urinary Na and UNaV in combined therapy was significantly higher which could be due to the full agonistic activity for PPAR-γ receptors, therefore, possesses renoprotective effects in diabetic SHRs. Urinary potassium and Na:K, serves as indirect marker of kidney function of aldosterone acting on the renal cortical collecting tubules [50], and can be taken as marker of ANG II level. Present study findings indicates that urinary potassium increased, whereas, Na:K significantly decreased as compared to their non-diabetic SHRs which is suggestive of increased ANG II level, which parallels with previous reports of Sowers [51]. Moreover, decreased Na:K is suggestive of increased aldosterone activity and supports fact that persistent hyperglycemia stimulates the ANG and renin expression in tubular and mesangial cells [42]. We observed that diabetic rats treated with adiponectin and combination with pioglitazone reversed urinary potassium concentration and Na:K significantly in treated groups with more pronounced effects.

The most significant findings of our experiment during acute surgical phase is the blunted vasoactive responses of the renal vasculature to adrenergic agonists in diabetic SHRs treated with chronic administration of pioglitazone and adiponectin through exogenous routes. We observed that pioglitazone improved endothelial function in the diabetic SHRs by reducing the vasoactive responses to exogenously administered alpha adrenergic agonists probably via increased plasma adiponectin levels through NO production. The diabetic SHRs expressed increased renal vasoconstrictor responses to ANG II and adrenergic agonists relative to normotensive rats, whereas, adiponectin and combination of adiponectin with pioglitazone treatments, the responses to adrenergic agonists were attenuated to a greater extent versus independent treatments. This indicates that adiponectin acts to depress signaling after adrenoceptors stimulation within the kidney vasculature, whereas, adiponectin exerts its effect through adipo R1 and adipo R2 receptors, which are mainly present in the kidneys [52]. The vasoactive responses in adiponectin treated groups also blunted alpha adrenergic agonists, but to lesser extent as compared to combination with PPAR-γ agonist (pioglitazone) treated groups, whereas responses to blunt ANGII were more in adiponectin treated group as compared to pioglitazone treatment. The higher Ang II level might be responsible for the altered vascular activity, therefore, reduced responsiveness to exogenously given Ang II observed during the acute phase of the experiment. In experimental diabetes, Ang II (AT1) receptor expression is up regulated and Ang II (AT2) receptor is down regulated [53], whereas, PPAR-γ agonists (TZDs) have been proven to down regulate the expression of angiotensin AT1 receptor mRNA in vascular smooth muscle (VSMC) [54]. Adiponectin treated groups decreased the magnitude of renal vascular responses to both ANG II and alpha adrenergic agonists, which may be related to the inhibitory effects of adiponectin on sympathetic nerve activity, as it is believed to be present in cerebrospinal fluid [55], and its administration through central regulation reduces B.P and HR [56]. Moreover, adiponectin treatment proposes a relationship existence between AT1, α1-adrenoceptors and adiponectin receptors (adipo R1 and adipoR2), in the renal vasculature of diabetic SHRs during attenuation of the vasoconstrictor responses to ANG II and alpha adrenergic agonists administration. As hyperinsulinemia is a known

stimulus of SNS, lower insulin levels with pioglitazone associated with reduced sympathetic drive [57] and the ability of TZDs to interfere with RAAS plays a significant role [58]. A cross-talk relationship between SNS and RAAS have been proven as α1-adrenoceptors are down regulated during higher levels of sympathetic nervous activity [59], whereas, AT1 and α1-adreno-ceptors interact with each other in modulating adrenergically and Ang II-induced vasoconstriction in the renal vasculature of rats [60]. Our current study findings demonstrate that full PPAR-γ agonist (pioglitazone) and adiponectin have interactions with RAAS at multiple levels although the exact level of interaction may vary across both diabetic and hypertensive states. Therefore, we may conclude from the findings of this study that renal vasculature of diabetic SHRs exhibit an interactive cross-talk relationship among alpha adrenoceptors, PPAR-γ and adiponectin receptors (adipo R1 and adipoR2).

## Conclusions

1. Adiponectin receptors and PPAR-γ has a considerable function in the control of renal haemodynamics in diabetic SHRs.

2. Adiponectin reduces blood pressure and improves renal haemodynamics in SHRs.

3. A degree of synergism exists between adiponectin and pioglitazone (PPAR-γ agonist).

4. A cross-talk relationship exists between adiponectin receptors, PPAR-γ and alpha adrenoceptors in renal vasculature of diabetic SHRs.

5. Adiponectin possesses renoprotective effects in diabetic SHRs by attenuating renal vascular reactivity through eNOs stimulation, thus maintains normal renal physiology.

## Supporting information

**S1 Table.**
(DOCX)

**S2 Table.**
(DOCX)

**S3 Table.**
(PDF)

**S1 File.**
(DOCX)

**S2 File.**
(DOCX)

**S3 File.**
(DOCX)

## Author Contributions

**Conceptualization:** Sheryar Afzal.

**Data curation:** Sheryar Afzal, Olorunfemi A. Eseyin.

**Formal analysis:** Sheryar Afzal, Olorunfemi A. Eseyin.

**Investigation:** Sheryar Afzal.

**Methodology:** Sheryar Afzal.

**Project administration:** Munavvar Abdul Sattar.

**Resources:** Munavvar Abdul Sattar.

**Supervision:** Munavvar Abdul Sattar, Edward James Johns.

**Validation:** Munavvar Abdul Sattar.

**Visualization:** Edward James Johns, Olorunfemi A. Eseyin.

**Writing – original draft:** Sheryar Afzal.

**Writing – review & editing:** Sheryar Afzal, Edward James Johns.

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
