## [Decision Letter · Decision Letter 0]

1 Apr 2020

PONE-D-20-04224

Renoprotective and haemodynamic effects of adiponectin and peroxisome proliferator-activated receptor agonist, pioglitazone, in renal vasculature of diabetic Spontaneously hypertensive rats

PLOS ONE

Dear Dr. Afzal,

Thank you for submitting your manuscript to PLOS ONE. After careful consideration, we feel that it has merit but does not fully meet PLOS ONE’s publication criteria as it currently stands. Therefore, we invite you to submit a revised version of the manuscript that addresses, experimentally, if necessary, the points raised by reviewer #2 during the review process.

We would appreciate receiving your revised manuscript by May 16 2020 11:59PM. To enhance the reproducibility of your results, we recommend that if applicable you deposit your laboratory protocols in protocols.io, where a protocol can be assigned its own identifier (DOI) such that it can be cited independently in the future. For instructions see: http://journals.plos.org/plosone/s/submission-guidelines#loc-laboratory-protocols

We look forward to receiving your revised manuscript.

Kind regards,

Luis Eduardo M Quintas, Ph.D.

Academic Editor

PLOS ONE

Journal Requirements:

Reviewers' comments:

Reviewer's Responses to Questions

**Comments to the Author**

1. Is the manuscript technically sound, and do the data support the conclusions?

Reviewer #1: Yes

Reviewer #2: No

2. Has the statistical analysis been performed appropriately and rigorously? 

Reviewer #1: Yes

Reviewer #2: Yes

3. Have the authors made all data underlying the findings in their manuscript fully available?

Reviewer #1: Yes

Reviewer #2: No

4. Is the manuscript presented in an intelligible fashion and written in standard English?

Reviewer #1: Yes

Reviewer #2: Yes

5. Review Comments to the Author

Reviewer #1: Earlier studies have shown that plasma adiponectin concentration is reduced in patients with hypertension and renal dysfunctions. In this manuscript the authors have reported the results of their study exploring the effects of pioglitazone which serves as an agonist of adiponectin and peroxisome proliferator-activated receptor (PPAR-γ), on renal and vascular system in diabetes-induced SHR rats. Their study showed that adiponectin exerts renal protective effects and improves renal hemodynamics through adiponectin receptors and PPAR-γ in diabetic SHR and point to the synergism between adiponectin and pioglitazone.

The study is well designed and executed, the results are interesting, and the paper is well written

Reviewer #2: Manuscript PONE-D-20-04224 entitled

Renoprotective and haemodynamic effects of adiponectin and peroxisome proliferator-activated receptor agonist, pioglitazone, in renal vasculature of diabetic Spontaneously hypertensive rats

Reviewers' comments:

The manuscript " Renoprotective and haemodynamic effects of adiponectin and peroxisome proliferator-activated receptor agonist, pioglitazone, in renal vasculature of diabetic Spontaneously hypertensive rats" it is an interesting research; the authors shows some benefit of pioglitazone combined with adiponectin in some parameters such as mean arterial blood pressure and renal vascular reactivity. The study design is OK and tests well performed and some interesting results are shown, however, the discussion section is very speculative and the conclusions are not supported by the results.

Major comments

1.The lack of water and food intake data make it very hard to interpret the results ?

2. The lack of real GFR (since diabetic rats face severe proteolysis the use of the creatinine ratio oos not a good predictor), filtration fraction and medullar blood flow (wich is not autoregulated and the cortical renal blood flow) also difficults the interpretations of the results;

3. Both hypertension and diabetes have increased urinary albumin/creatinine as a hallmark, it would be a better marker than Na/K ratio as a measure of renoprotection. Indeed, no renal marker or histopathological finding support this renoprotective claim of the authors;

4. The authors discuss about eNOs, Na/KATPase, sympathetic activity, renin levels, aldosterone levels, etc but did not actually measure any of these parameters;

5. for instance, its very difficult to discuss the data since the authors infuse 4 different agonists both using 3 different doses and we don’t have any data in renal sympathetic activity, renin levels, ACE activity, etc;

6. UNaV, FENa are not hemodynamic parameters and should not be considered as hemodynamic data in this manuscript;

7. The authors claim that adiponectin induces natriuresis and diuresis but this data is not shown in the present manuscript;

Minor comments:

1. 2 ml of blood sampling seems too excessive, it is there a good reason for such a high volume of blood taken from this already critic diabetic animals??

2.NA/PE/ME are cited without any previous description. Whats the rationale to use these drugs all together in one animal. Why not to choose only one or at most two?

3. 3 days of acclimatization period seems to be a short period of adaptation and in this condition food and water intake should have been reported, especially because potassium excretions is very sensitive to diet.

4. There are several typing errors and English grammar errors that must be corrected.

6. PLOS authors have the option to publish the peer review history of their article (what does this mean?). If published, this will include your full peer review and any attached files.

Reviewer #1: Yes: ND Vaziri MD,MACP

Reviewer #2: No

---

## [Author Response · Author response to Decision Letter 0]

9 Jun 2020

Dear Academic Editor, Luis Eduardo M Quintas, With reference to manuscript ID PONE-D-20-04224R1 and Plos One Decision, many thanks to your kind self and PLOS ONE team to grant me an opportunity to resubmit my manuscript for publication purpose entitled “Renoprotective and haemodynamic effects of adiponectin and peroxisome proliferator-activated receptor agonist, pioglitazone, in renal vasculature of diabetic Spontaneously hypertensive rats” after major changes. Kindly note that the manuscript has been comprehensively and adequately addressed and revised as per the comments raised by the reviewers and journal requirements including the file and author templates of the journal. Moreover, the complete data sets for the results as underlying minimal data availability, to reach conclusion and to replicate the reported study, has been uploaded as supporting information files for all the parameters measured in the study as per journals requirements and directions. A rebuttal letter responding to the reviewer’s comments has been uploaded in a separate files comprehensively addressing to all the points raised by the reviewer named “Response to Reviewers with a marked –up copy highlighting changes as required after the comments besides an unmarked version of the Manuscript.

---

## [Decision Letter · Decision Letter 1]

7 Jul 2020

PONE-D-20-04224R1

Renoprotective and haemodynamic effects of adiponectin and peroxisome proliferator-activated receptor agonist, pioglitazone, in renal vasculature of diabetic Spontaneously hypertensive rats

PLOS ONE

Dear Dr. Afzal,

Thank you for submitting your manuscript to PLOS ONE. After careful consideration, we feel that it has merit but does not fully meet PLOS ONE’s publication criteria as it currently stands. Therefore, we invite you to submit a revised version of the manuscript that addresses the points raised during the review process.

We look forward to receiving your revised manuscript.

Kind regards,

Luis Eduardo M Quintas, Ph.D.

Academic Editor

PLOS ONE

Reviewers' comments:

Reviewer's Responses to Questions

**Comments to the Author**

1. If the authors have adequately addressed your comments raised in a previous round of review and you feel that this manuscript is now acceptable for publication, you may indicate that here to bypass the “Comments to the Author” section, enter your conflict of interest statement in the “Confidential to Editor” section, and submit your "Accept" recommendation.

Reviewer #2: (No Response)

2. Is the manuscript technically sound, and do the data support the conclusions?

Reviewer #2: Partly

3. Has the statistical analysis been performed appropriately and rigorously? 

Reviewer #2: Yes

4. Have the authors made all data underlying the findings in their manuscript fully available?

Reviewer #2: Yes

5. Is the manuscript presented in an intelligible fashion and written in standard English?

Reviewer #2: Yes

6. Review Comments to the Author

Reviewer #2: The authors addressed some of the comments but the main core of the questions raised continued unanswered. The renal functions depends in a great deal to water and food intake especially potassium excretion. True hemodynamic parameters such as GFR, filtration fraction, medular blood flow, etc were not used. True markers of renal integrity were not used also neither morphological analysis were performed what difficults interpretation of the data and weakens the conclusions of the authors. The pharmacological study was also not weel designed since 3 drugs in different doses were used in the same animal and no paramenter of compensations (such as sympathetic acitivity, ACE, renin, aldosterone,etc) were measured tu support conclusions. On the top of that no mechanism to explain the claimed benefits of adiponectin and piaglitazone claime renoprotective effect.

7. PLOS authors have the option to publish the peer review history of their article (what does this mean?). If published, this will include your full peer review and any attached files.

Reviewer #2: No

---

## [Author Response · Author response to Decision Letter 1]

21 Aug 2020

Many thanks to the Editor and PLOS ONE team to grant me an opportunity to resubmit my manuscript for publication purpose after major changes. Many thanks to reviewer for acknowledging the comments in the previous version of the manuscripts as “responses to reviewer comments”, as we tried our best to address the queries asked asked by the reviewer, and highlighting the remaining points. These required points raised by the reviewer have been comprehensively readdressed as per data availability and study protocols. Comments have been attached in a separate file named "Response to Reviewer" with an understanding for the final acceptance of the manuscript. With apologies for the delay in response on account of hectic academic activities. With regards and thanks to Plos One team and Editor in Chief.

---

## [Decision Letter · Decision Letter 2]

15 Oct 2020

Renoprotective and haemodynamic effects of adiponectin and peroxisome proliferator-activated receptor agonist, pioglitazone, in renal vasculature of diabetic Spontaneously hypertensive rats

PONE-D-20-04224R2

Dear Dr. Afzal,

We’re pleased to inform you that your manuscript has been judged scientifically suitable for publication and will be formally accepted for publication once it meets all outstanding technical requirements.

Kind regards,

Luis Eduardo M Quintas, Ph.D.

Academic Editor

PLOS ONE

Additional Editor Comments (optional):

Reviewers' comments:

Reviewer's Responses to Questions

**Comments to the Author**

1. If the authors have adequately addressed your comments raised in a previous round of review and you feel that this manuscript is now acceptable for publication, you may indicate that here to bypass the “Comments to the Author” section, enter your conflict of interest statement in the “Confidential to Editor” section, and submit your "Accept" recommendation.

Reviewer #2: All comments have been addressed

2. Is the manuscript technically sound, and do the data support the conclusions?

Reviewer #2: Yes

3. Has the statistical analysis been performed appropriately and rigorously? 

Reviewer #2: Yes

4. Have the authors made all data underlying the findings in their manuscript fully available?

Reviewer #2: Yes

5. Is the manuscript presented in an intelligible fashion and written in standard English?

Reviewer #2: Yes

6. Review Comments to the Author

Reviewer #2: The authors have revised the papers and included new data that made this paper more interesting and the cunclusions are now better supported by the data presented.

7. PLOS authors have the option to publish the peer review history of their article (what does this mean?). If published, this will include your full peer review and any attached files.

Reviewer #2: No

---

## [Editor Report · Acceptance letter]

30 Oct 2020

PONE-D-20-04224R2 

Renoprotective and haemodynamic effects of adiponectin and peroxisome proliferator-activated receptor agonist, pioglitazone, in renal vasculature of diabetic Spontaneously hypertensive rats  

Dear Dr. Afzal:

I'm pleased to inform you that your manuscript has been deemed suitable for publication in PLOS ONE. Congratulations! Your manuscript is now with our production department. 

Kind regards, 

on behalf of

Dr. Luis Eduardo M Quintas 

Academic Editor

PLOS ONE